# PLANNING IMMEDIATE LANDMARKS OF TARGETS FOR MODEL-FREE SKILL TRANSFER ACROSS AGENTS

## ABSTRACT

In reinforcement learning applications, agents usually need to deal with various input/output features when specified with different state and action spaces by their developers or physical restrictions, indicating re-training from scratch and considerable sample inefficiency, especially when agents follow similar solution steps to achieve tasks. In this paper, we aim to transfer pre-trained skills to alleviate the above challenge. Specifically, we propose PILoT, i.e., Planning Immediate Landmarks of Targets. PILoT utilizes the universal decoupled policy optimization to learn a goal-conditioned state planner; then, we distill a goal-planner to plan immediate landmarks in a model-free style that can be shared among different agents. In our experiments, we show the power of PILoT on various transferring challenges, including few-shot transferring across action spaces and dynamics, from low-dimensional vector states to image inputs, from simple robot to complicated morphology; and we also illustrate PILoT provides a zero-shot transfer solution from a simple 2D navigation task to the harder Ant-Maze task.

## 1 INTRODUCTION

Recent progress of Reinforcement Learning (RL) has promoted considerable developments in resolving kinds of decision-making challenges, such as games (Guan et al., 2022), robotics (Gu et al., 2017) and even autonomous driving (Zhou et al., 2020). However, most of these work are designed for a single task with a particular agent. Recently, researchers have developed various goal-conditioned reinforcement learning (GCRL) methods in order to obtain a generalized policy to settle a group of homogeneous tasks with different goals simultaneously (Liu et al., 2022a), but are still limited in the same settings of environment dynamics/reward, and the same state/action space of the agent. Many existing solutions in the domain of Transfer RL (Zhu et al., 2020) or Meta RL (Yu et al., 2020) aim to transfer among different dynamics/reward with the same agent, but care less for the shared knowledge across agents with different state/action spaces.

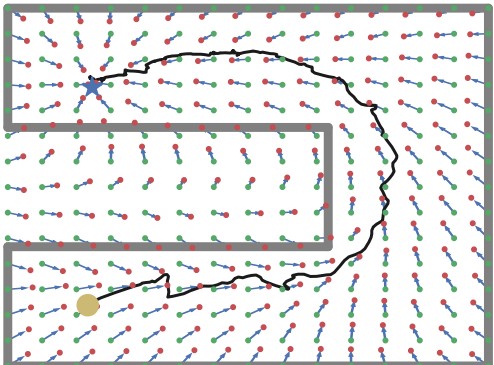

Figure 1: Zero-shot transferring on Ant-Maze, where the Ant agent starts from the yellow point to the desired goal (big blue star). PILoT provide planned immediate landmarks (small red points) given the temporal goal (green points) and the desired goal (small blue point), learned from a naive 2D maze task.

There are many motivations and scenarios encouraging us to design a transferring solution among agents: a) deployed agents facing with changed observing features, for instance, non-player characters (NPC) trained and updated for incremental scenes of games (Juliani et al., 2018), robots with new sensors due to hardware replacement (Bohez et al., 2017); b) agents in different morphology have to finish the same tasks (Gupta et al., 2021), such as run a complicate quadruped robotic following a much simpler simulated robot (Peng et al., 2018); c) improving the learning efficiency with rich and redundant observations or complicate action spaces, like transferring the knowledge from compact low-dimensional vector input to high-dimensional image features (Sun et al., 2022).

Some previous works have made progress on transfer across agents on a single task. (Sun et al., 2022) transferred across different observation spaces with structural-similar dynamics and the same action spaces via learning a shared latent-space dynamics to regularize the policy training. On the other hand, (Liu et al., 2022b) decouples a policy as a state planner that predicts the consecutive target state, and an inverse dynamics model that delivers action to achieve the target state, which allows transferring across different action spaces and action dynamics, but limit in the same state space and state transitions.

In this paper, we propose a more general solution for transferring the multi-task skills across agents with heterogeneous action spaces and observation space, named Planning Immediate Landmarks of Targets (PILoT). Our method works under the assumption that agents share the same goal transition to finish tasks, but without any prior knowledge of the inter-task mapping between the different state/action spaces, and agents can not interact with each other.

The whole workflow of PILoT is composed of three stages, including pre-training, distillation and transfer: 1) the *pre-training stage* extends the decoupled policy to train a universal state planner on simple tasks with universal decoupled policy optimization; 2) the *distillation stage* distills the knowledge of state planner into an immediate goal planner, which is then utilized to 3) the *transferring stage* that plans immediate landmarks in a model-free style serving as dense rewards to improve the learning efficiency or even straightforward goal guidance. Fig. 1 provides a quick overview of our algorithm for zero-shot transferring on Ant-Maze. Correspondingly, we first train a decoupled policy on a simple 2D maze task to obtain a universal state planner, then distill the knowledge into a goal planner that predicts the immediate target goal (red points) to reach given the desired goal (blue point) and arbitrary started goal (green points). Following the guidance, the Ant controllable policy is pre-trained on a free ground without the walls can be directly deployed on the maze environment without any training. As the name suggests, we are providing immediate landmarks to guide various agents like the runway center line light on the airport guiding the flight to take off.

Comprehensive challenges are designed to examine the superiority of PILoT on the skill transfer ability, we design a set of hard transferring challenges, including few-shot transfer through different action spaces and action dynamics, from low-dimensional vectors to image inputs, from simple robots to complicated morphology, and even zero-shot transfer. The experimental results present the learning efficiency of PILoT transferred on every tasks by outperforming various baseline methods.

## 2 PRELIMINARIES

**Goal-Augmented Markov Decision Process.** We consider the problem of goal-conditioned reinforcement learning (GCRL) as a $\gamma$-discounted infinite horizon goal-augmented Markov decision process (GA-MDP) $\mathcal{M} = \langle \mathcal{S}, \mathcal{A}, \mathcal{T}, \rho_0, r, \gamma, \mathcal{G}, p_g, \phi \rangle$, where $\mathcal{S}$ is the set of states, $\mathcal{A}$ is the action space, $\mathcal{T} : \mathcal{S} \times \mathcal{A} \times \mathcal{S} \to [0, 1]$ is the environment dynamics function, $\rho_0 : \mathcal{S} \to [0, 1]$ is the initial state distribution, and $\gamma \in [0, 1]$ is the discount factor. The agent makes decisions through a policy $\pi(a|s) : \mathcal{S} \to \mathcal{A}$ and receives rewards $r : \mathcal{S} \times \mathcal{A} \to \mathbb{R}$, in order to maximize its accumulated reward $R = \sum_{t=0}^{t} \gamma^t r(s_t, a_t)$. Additionally, $\mathcal{G}$ denotes the goal space w.r.t tasks, $p_g$ represents the desired goal distribution of the environment, and $\phi : \mathcal{S} \to \mathcal{G}$ is a tractable mapping function that maps the state to a specific goal. One typical challenge in GCRL is reward sparsity, where usually the agent can only be rewarded once it reaches the goal:

$$r_g(s_t, a_t, g) = \mathbb{1}(\text{the goal is reached}) = \mathbb{1}(\|\phi(s_{t+1}) - g\| \leq \epsilon) . \tag{1}$$

Therefore, GCRL focuses on multi-task learning where the task variationality comes only from the difference of the reward function under the same dynamics. To shape a dense reward, a straightforward idea is to utilizing a distance measure $d$ between the achieved goal and the final desired goal, *i.e.*, $\tilde{r}_g(s_t, a_t, g) = -d(\phi(s_{t+1}), g)$. However, this reshaped reward will fail when the agent must first increase the distance to the goal before finally reaching it, especially when there are obstacles on the way to the target (Trott et al., 2019).

In our paper, we work on a deterministic environment dynamics function $\mathcal{T}$, such that $s' = \mathcal{T}(s, a)$, and we allow redundant actions, *i.e.*, the transition probabilities can be written as linear combination of other actions'. Formally, there exists of a state $s_m \in \mathcal{S}$, an action $a_n \in \mathcal{A}$ and a distribution $p$ defined on $\mathcal{A} \setminus \{a_n\}$ such that $\int_{\mathcal{A} \setminus \{a_n\}} p(a) \mathcal{T}(s'|s_m, a) \, \mathrm{d}a = \mathcal{T}(s'|s_m, a_n)$.

**Decoupled Policy Optimization** Classical RL methods learn a state-to-action mapping policy function, whose optimality is ad-hoc to a specific task. In order to free the agent to learn a high-level planning strategy that can be used for transfer, Liu et al. (2022b) proposed Decoupled Policy Optimization (DePO) which decoupled the policy structure by a state transition planner and an inverse dynamics model as:

$$\pi(\cdot|s) = \int_{s'} h_\pi(s'|s) I(\cdot|s, s') \, \mathrm{d}s' = \mathbb{E}_{\hat{s}' \sim h_\pi(\hat{s}'|s)} \Big[ I(\cdot|s, \hat{s}') \Big] . \tag{2}$$

To optimize the decoupled policy, DePO first optimizes the inverse dynamics model via supervised learning, and then performs policy gradient assuming a fixed but locally accurate inverse dynamics function. DePO provides a way of planning without training an environment dynamics model. The state planner of DePO pre-trained on simple tasks can be further transferred to agents with various action spaces or dynamics. However, as noted below, the transferring ability of DePO limits in the same state space transition. In this paper, we aims to derive a more generalized skill transferring solution utilizing the common latent goal space shared among tasks and agents.

## 3 Transfer across Agents

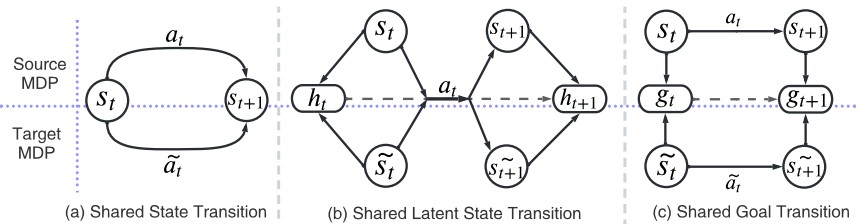

(a) Shared State Transition    (b) Shared Latent State Transition    (c) Shared Goal Transition

Figure 2: Comparison of the different assumption for transferring across agents of previous works (Liu et al., 2022b; Sun et al., 2022) and PILoT. (a) (Liu et al., 2022b) allows for transferring across action spaces and but ask the same state space and state transition. (b) (Sun et al., 2022) transfers across different state spaces but require there exists a shared latent state space and dynamics. (c) PILoT provides a generalized transferring ability for both state space and action space, but asks for a shared underlying latent goal transition.

In this section, we explain the problem setup of transfer across agents. Formally, we pre-train and learn knowledge from a source GA-MDP $\mathcal{M} = \langle \mathcal{S}, \mathcal{A}, \mathcal{T}, \rho_0, r, \gamma, \mathcal{G}, p_g, \phi \rangle$ and transfer to a target GA-MDP $\tilde{\mathcal{M}} = \langle \tilde{\mathcal{S}}, \tilde{\mathcal{A}}, \tilde{\mathcal{T}}, \tilde{\rho}_0, \tilde{r}, \gamma, \mathcal{G}, p_g, \tilde{\phi} \rangle$. Here we allow significant difference between the state spaces $\mathcal{S}, \tilde{\mathcal{S}}$, and action spaces $\mathcal{A}, \tilde{\mathcal{A}}$. Therefore, both the input / output shapes of the source policy are totally different from the target one and therefore it is challenging to transfer a shared knowledge.

To accomplish the objective of transfer, prior works always make assumptions about the shared structure (Fig. 2). For example, (Sun et al., 2022) proposed to transfer across significantly different state spaces with the same action space and similar structure between dynamics, *i.e.*, a mapping between the source and target state spaces exists such that the transition dynamics shares between two tasks under the mapping. In comparison, (Liu et al., 2022b) pay attention on transferring across action spaces under the same state space and the same state transitions, *i.e.*, a action mapping between the source and target exists such that the transition dynamics shares between two tasks under the mapping. In this paper, we take a more general assumption by only require agents have a shared goal transition, and allow transferring across different observation spaces and action spaces. We argue that this is a reasonable requirement in real world since for tasks like robot navigation, different robot agents can share a global positioning system constructed by techniques like SLAM that allows them to quickly figure out their 3D position in the world. Formally, the assumption corresponds to:

**Assumption 1.** *There exists a function $f : \mathcal{A} \to \tilde{\mathcal{A}}$ such that $\forall s, s' \in \mathcal{S}, \forall a \in \mathcal{A}$, and $\exists \tilde{s}, \tilde{s}' \in \tilde{\mathcal{S}}, \exists \tilde{a} \in \tilde{\mathcal{A}}$,*

$$\mathcal{T}(s'|s, a) = \tilde{\mathcal{T}}(\tilde{s}'|\tilde{s}, f(a)), r(s, a, g^t) = \tilde{r}(\tilde{s}, \tilde{a}, g^t), \phi(s) = \tilde{\phi}(\tilde{s}), \phi(s') = \tilde{\phi}(\tilde{s}') .$$

Here $\phi$ is usually a many-to-one mapping, such as an achieved position or the velocity of a robot. $f$ can be any function, like many-to-one mapping, where several target actions relate to the same

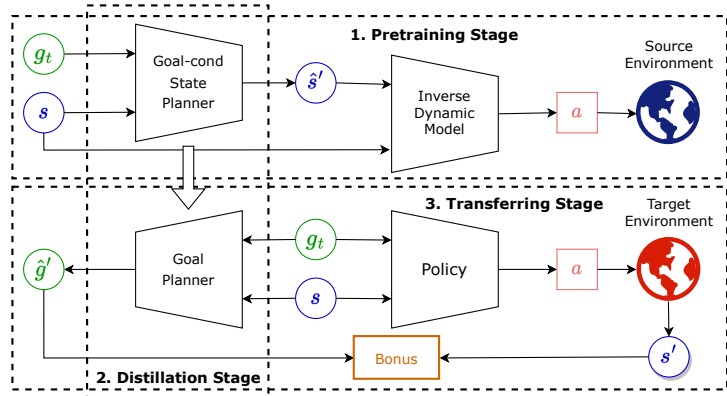

Figure 3: Overview of universal decoupled policy optimization (PILoT) framework. PILoT leverages a transferring-by-pre-training process, including the stage of pre-training, distillation and transferring.

source action; or one-to-many mapping; or non-surjective, where there exists a source action that does not correspond to any target action.

## 4 PLANNING IMMEDIATE LANDMARKS OF TARGETS

In this section, we introduce our generalized multi-task skills transfer solution with the proposed Planning Immediate Landmarks of Targets (PILoT) framework. First, we demonstrate how we derive the training procedure of a universal decoupled policy structure for multiple tasks in the pre-training stage; then, we characterize the distillation stage which obtain a goal planner for providing the informative reward bonus or zero-shot guidance for the transferring stage. An overview of the method is shown in Fig. 3, and we list the step-by-step algorithm in Algo. 1.

### 4.1 UNIVERSAL DECOUPLED POLICY OPTIMIZATION

In order to derive a generally transferable solution, we first extend the decoupled policy structure (Liu et al., 2022b) into a goal-conditioned form. Formally, we decouple the goal-conditioned policy $\pi(a|s, g^t)$ as:

$$\pi(a|s, g^t) = \int_{s'} h_\pi(s'|s, g^t) I(\cdot|s, s') \, \mathrm{d}s' = \mathbb{E}_{\hat{s}' \sim h_\pi(\hat{s}'|s, g^t)} \Big[ I(\cdot|s, \hat{s}') \Big] , \quad (3)$$

where $g^t$ is the target goal, $h_\pi$ is a goal-conditioned state planner. Approximating the planner by neural networks (NNs), we can further apply the reparameterization trick and bypass explicitly computing the integral over $s'$ as

$$s' = h(\epsilon; s, g^t), \quad \pi(a|s, g^t) = \mathbb{E}_{\epsilon \sim \mathcal{N}} \left[ I(a|s, h(\epsilon; s, g^t)) \right] , \quad (4)$$

where $\epsilon$ is an input noise vector sampled from some fixed distribution, like a Gaussian. The inverse dynamics $I$ should serve as a control module known in advance for reaching the target predicted by the planner. When it must be learned from scratch, we can choose to minimize the divergence (for example, KL) between the inverse dynamics of a sampling policy $\pi_\mathcal{B}$ and the $\phi$-parameterized function $I_\phi$, i.e.,

$$\min_\psi L^I = \mathbb{E}_{(s,s') \sim \pi_\mathcal{B}} [\mathrm{D}_\mathrm{f}(I_{\pi_\mathcal{B}}(a|s, s') \| I_\phi(a|s, s'))] . \quad (5)$$

It is worth noting that the model is only responsible and accurate for states encountered by the current policy instead of the overall state space. In result, the inverse dynamics model is updated every time before updating the policy.

To update the decoupled policy, particularly, the goal-conditioned state planner $h_\pi$, given that the inverse dynamics to be an accurate local control module for the current policy and the inverse dynamics function $I$ is static when optimizing the policy function, we adopt the decoupled policy gradient (DePG) as derived in (Liu et al., 2022b):

$$\nabla_\psi \mathcal{L}^\pi = \mathbb{E}_{(s,a) \sim \pi, g^t \sim p_g, \epsilon \sim \mathcal{N}} \left[ \frac{Q(s,a,g)}{\pi(a|s, g^t)} \big( \nabla_h I(a|s, h_\psi(\epsilon; s, g^t)) \nabla_\psi h_\psi(\epsilon; s, g^t) \big) \right] , \quad (6)$$

which can be seen as taking the knowledge of the inverse dynamics about the action $a$ to optimize the planner by a prediction error $\Delta s' = \alpha \nabla_h I(a|s, h(\epsilon; s))$ where $\alpha$ is the learning rate. However, (Liu et al., 2022b) additionally pointed that there exists the problem of agnostic gradients that the optimization direction is not always lead to a legal state transition. To alleviate the challenge, they proposed calibrated decoupled policy gradient to ensure the state planner from predicting a infeasible state transition. In this paper, we turn to a simpler additional objective for constraints, *i.e.*, we maximize the probability of predicting the legal transitions that are sampled by the current policy, which is demonstrated to have a similar performance in (Liu et al., 2022b):

$$\max \mathbb{E}_{(s,s')\sim\pi}[h(s'|s, g^t)] , \tag{7}$$

Therefore, the overall gradient for updating the planner becomes:

$$\nabla_\psi \mathcal{L}^\pi = \mathbb{E}_{(s,a,s')\sim\pi, g^t\sim p_g, \epsilon\sim\mathcal{N}} \left[ \frac{Q(s,a,g)}{\pi(a|s,g^t)} \big( \nabla_h I(a|s, h_\psi(\epsilon; s, g^t)) \nabla_\psi h_\psi(\epsilon; s, g^t) \big) + \lambda \nabla_\psi h_\psi(\epsilon; s, g^t) \right] , \tag{8}$$

where $\lambda$ is the hyperparamer for trading off the constraint. Note that such a decoupled learning scheme also allows incorporating various relabeling strategy to further improving the sample efficiency, such as HER (Andrychowicz et al., 2017).

## 4.2 Goal Planner Distillation

In order to transfer the knowledge to new settings, we leverage the shared latent goal space and distill a goal planner from the goal-conditioned state planner, *i.e.*, we want to predict the consecutive goal given the current goal and the target goal. Formally, we aims to obtain a $\omega$-parameterized function $f_\omega(g'|g, g^t)$, where $g'$ is the next goal to achieve, $g = \phi(s)$ is the current goal the agent is achieved, and $g^t$ is the target. This can be achieved by treating the state planner $h_\psi(s'|s, g^t)$ as the teacher, and $f_\omega(g'|g, g^t)$ becomes the student. The objective of the distillation is constructed as an MLE loss:

$$\nabla_\omega \mathcal{L}^f = \max_\omega \mathbb{E}_{s\sim\mathcal{B}, \tilde{s}'\sim h_\psi g^t\sim p_g}[f_\omega(\tilde{g}'|g, g^t)], \text{ where } g = \phi(s), \tilde{g}' = \phi(\tilde{s}') , \tag{9}$$

where $\mathcal{B}$ is the replay buffer, $\phi$ is the mapping function that translates a state to a specific goal. With the distilled goal planner, we can now conduct goal planning without training and querying an environment dynamics model as in Zhu et al. (2021); Chua et al. (2018).

## 4.3 Transfer Multi-Task Knowledge across Agents

A typical challenge for GCRL is the rather sparse reward function, and simply utilizing the Euclid distance of the final goal and the current achieved goal can lead to additional sub-optimal problems. To this end, with PILoT having been distilled for acquiring plannable goal transitions, it is natural to construct a reward function (or bonus) leveraging the difference between the intermediate goals to reach and the goal that actually achieves for transferring. In particular, when the agent aims to go to $g_t$ with the current achieved goal $g$, we exploit the distilled planner $f_\omega$ from PILoT to provide reward as similarity of goals:

$$r(s, a, \hat{g}') = \frac{\phi(s') \cdot \hat{g}'}{\|\phi(s')\|\|\hat{g}'\|}, \text{ where } s' = \mathcal{T}(s, a) , \hat{g}' \sim f_\omega(\hat{g}'|g, g^t) . \tag{10}$$

Note that we avoid the different scale problem among different agents by using the form of cosine distance. Thereafter, we can actually transfer to a totally different agent. For example, we can learn a locomotion task from a easily controllable robot, and then transfer the knowledge to a complex one with more joints which is hard to learn directly from the sparse rewards; or we can learn from a low-dimensional ram-based agent and then transfer on high-dimensional image inputs. To verify the effectiveness of PILoT, we design various of transferring settings in Section 6.

## 5 Related Work

**Goal-conditioned RL.** Our work lies in the formulation of goal-conditioned reinforcement learning (GCRL). The existence of goals, which can be explained as skills, tasks, or targets, making it possible for our skill transfer across various agents with different state and action space. In the

literature of GCRL, researchers focus on alleviating the challenges in learning efficiency and generalization ability, from the perspective of optimization Trott et al. (2019); Ghosh et al. (2021); Zhu et al. (2021), generating or selecting sub-goals Florensa et al. (2018); Pitis et al. (2020) and relabeling Andrychowicz et al. (2017); Zhu et al. (2021). A comprehensive GCRL survey can be further referred to Liu et al. (2022a). In these works, the goal given to the policy function is either provided by the environment or proposed by a learned function. In comparison, in our paper, the proposed UDPO algorithm learns the next target states in an end-to-end manner which can be further distilled into a goal planner that can be used to propose the next target goals.

**Hierarchical reinforcement learning.** The framework of UDPO resembles Hierarchical Reinforcement Learning (HRL) structures, where the state planner plays like a high-level policy and the inverse dynamics as the low-level policy. Typical paradigm of HRL trains the high-level policy using environment rewards to predict sub-goals (or called options) that the low-level policy should achieve, and learn the low-level policy using handcrafted goal-reaching rewards to provide the action and interacts with the environment. Generally, most of works provided the sub-goals / options by the high-level policy are lied in a learned latent space (Konidaris & Barto, 2007; Heess et al., 2016; Kulkarni et al., 2016; Vezhnevets et al., 2017; Zhang et al., 2022), keeping it for a fixed timesteps (Nachum et al., 2018; Vezhnevets et al., 2017) or learn to change the option (Zhang & Whiteson, 2019; Bacon et al., 2017). On the contrary, Nachum et al. (2018) and Kim et al. (2021) both predicted sub-goals in the raw form, while still training the high-level and low-level policies with separate objectives. As for Nachum et al. (2018), they trained the high-level policy in an off-policy manner; and for Kim et al. (2021), which focused on goal-conditioned HRL tasks as ours, they sampled and selected specific landmarks according to some principles, and asked the high-level policy to learn to predict those landmarks. Furthermore, their only sampled a goal from the high-level policy for a fixed steps, otherwise using a pre-defined goal transition process. Like UDPO, Li et al. (2020) optimized the two-level hierarchical policy in an end-to-end way, with a latent skill fixed for $c$ timesteps. The main contribution of HRL works concentrate on improving the learning efficiency on complicated tasks, yet UDPO aims to obtain every next targets for efficient transfer.

**Transferable RL.** Before our work, there are a few works have investigated transferable RL. In this endeavor, Srinivas et al. (2018) proposed to transfer an encoder learned in the source tasks, which maps the visual observation to a latent representation. When transfer to target tasks / agents, the latent distance from the goal image to the current observation is used to construct an obstacles-aware reward function. To transfer across tasks, Barreto et al. (2017); Borsa et al. (2018) utilized success features based on strong assumptions about the reward formulation, that decouples the information about the dynamics and the rewards into separate components, so that only the relevant module need to be retrained when the task changes. In order to reuse the policy, Heess et al. (2016) learned a bi-level policy structure, composed of a low-level domain-general controller and high-level task-specific controller. They transfer the low-level controller on the target tasks while retraining the high-level one. On the other hand, Liu et al. (2022b) decoupled the policy as a state planner and an inverse dynamic model, and shown that the high-level state planner can be transferred to agents with different action spaces. For generalizing and transferring across modular robots' morphology, Gupta et al. (2017) tried learning invariant visual features. Wang et al. (2018) and Huang et al. (2020) both exploited graph structure present in the agent's morphology, and proposed specific structured policies for learning the correlations between agent's components, while Gupta et al. (2021) proposed a transformer-based policy to model the morphology representation. Theses works with such structured policies, although can generalize to agents with unseen morphology, limit in modular robots with specific representation of the state space. Similar to PILoT, Sun et al. (2022) proposed to transfer across agents with different state spaces, which is done with a latent dynamics model trained on the source task. On the target tasks, the pre-trained dynamics model is transferred as a model-based regularizer for improving the learning efficiency. In comparison, PILoT can do effectively multi-tasks skill transfer to agents with different structures, state spaces and action spaces. First, we separate the state spaces and action spaces in the decoupled policy structure, and the state predictor can be transferred to target tasks with different actions spaces. Furthermore, we distill a planner in the goal space, which allows for planning consecutive target goals as transferable skills to different agent structures and state spaces.

# 6 EXPERIMENTS

We conduct a set of transferring challenges to examine the skill transfer capacity of PILoT.

## 6.1 EXPERIMENTAL SETUPS

**Environments, tasks and agents.** a) *Fetch-Reach*. The agent controls the robot arm to reach a target position. b) *Fetch-Push*. The robotic arm is controlled to push a block to a target position. c) *Reacher*. Controling a multi-joint reacher robot to reach a target position. d) *Point-Locomotion*. The agent controls an point robot to move to a target position freely. e) *Ant-Locomotion*. The agent controls an ant robot to move to a target position freely. f) *2D-Maze*. The agent controls a mass point in a U-shape maze to reach a target position. g) *Ant-Maze*. The agent controls an ant robot to move to a target position in a U-shape maze.

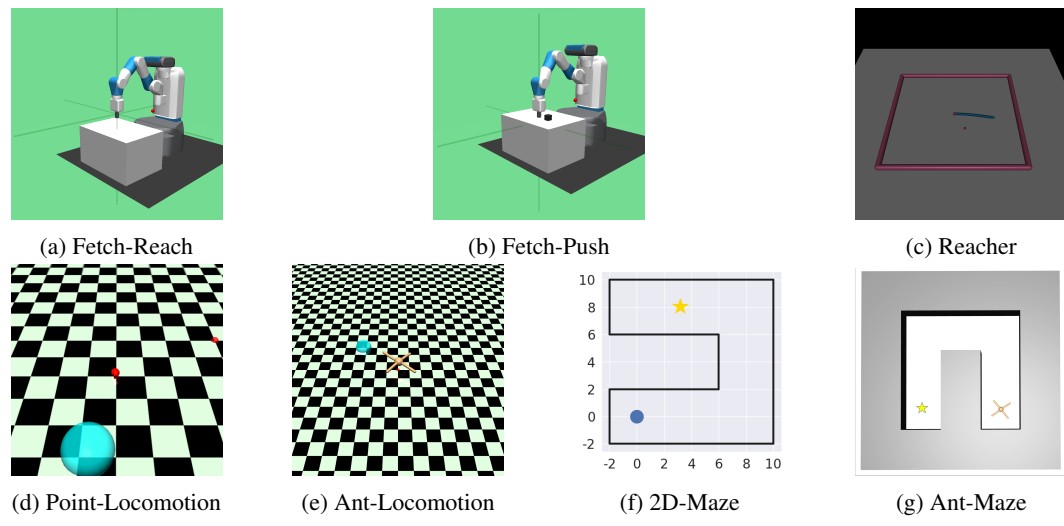

Figure 4: Illustration of tested environments.

**Designed transferring challenges.** 1) **Few-shot transfer to high-dimensional action spaces**: we take *Fetch-Reach* and *Fetch-Push* as the source tasks, and change the gravity, action dynamics and spaces to get the target tasks while sharing the same state space and state transition (see Appendix B.2 for details). 2) **Few-shot transfer to a complicate morphology**: we take *Point* robot with 6 observation dimensions as the source task, and *Ant* with 29 observation dimensions as the target task. 3) **Few-shot transfer from vector to image observations**: We take *Fetch-Reach* and *Reacher* with low-dimensional vector state as the source task, and the same environment with high-dimensional image observations as the target task. 4) **Zero-shot transfer**: we take *2D-Maze* and *Ant-Locomotion* as the source tasks, which are used to train a goal planner and a local controller separately; then we combine the knowledge of the goal planner and the local controller to master the task of *Ant-Maze* without any further training. All of these tasks varies in either in the state space or in the action space, but a pair of tasks must share the same goal space and transition, making it possible to transfer useful knowledge. In the sequal, we show different usages of PILoT on transferring such knowledge in different challenges.

**Implementation and baselines.** We choose several classical and recent representative works as our comparable baselines, for both source tasks and target tasks. For source tasks we want to compare the learning performance from scratch of the proposed UDPO algorithm with: i) **Hindsight Experience Replay (HER)** (Andrychowicz et al., 2017), a classical GCRL algorithm that trains goal-conditioned policy by relabeling the target goals as the samples appeared in future of the same trajectory for one particular state-action sample, which is also used as the basic strategy to train the UDPO policy; ii) **HIerarchical reinforcement learning Guided by Landmarks (HIGL)** Kim et al. (2021), an HRL algorithm that utilizes a high-level policy to propose landmark states for the

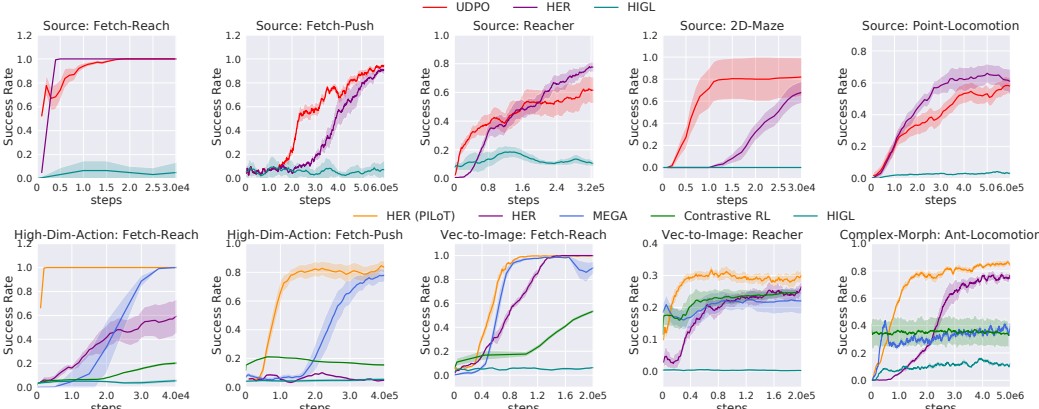

Figure 5: Training curves on four source tasks and four target tasks. `High-Dim-Action`: few-shot transfer to high-dimensional action space. `High-Dim-Action`: few-shot transferring to high-dimensional action space. `Vec-to-Image`: few-shot transferring from vector to image states. `Complex-Morph`: few-shot transfer to different morphology. UDPO denotes the learning algorithm proposed in Section 4.1. HER (PILoT) denotes HER with the transferring rewards provided by the proposed PILoT solution.

low-level policy to explore. As for the target tasks, we aim to show the transferring ability of PILoT from source tasks to target tasks, except HER and HIGL, we also compare a recent well-performed GCRL algorithm, **Maximum Entropy Gain Exploration (MEGA)** (Pitis et al., 2020), which proposed to enhance the exploration coverage of the goal space by sampling goals that maximize the entropy of past achieved goals; also, a very recent algorithm, **Contrastive RL** (Eysenbach et al., 2022), which design a contrastive representation learning solution for GCRL. Noted that contrastive RL requires the goal space the same as the state space (*e.g.*, both are images). For each baseline algorithm, we either take their suggested hyperparameters, or try our best to tune important ones.

In our transferring experiments, our PILoT solution first trains the decoupled policy by UDPO with HER's relabeling strategy for all source tasks; then, we distill the goal planner and generate dense reward signals to train a normal policy by HER in the target tasks, denoted as HER (PILoT).

## 6.2 RESULTS AND ANALYSIS

In the main context, we focus on presenting the training results on source tasks and the transferring results on target tasks. Additionally, we leave in depth analysis, ablation studies and hyperparameter choices in the Appendix C.

**Learning in the source tasks.** We first train UDPO on four source tasks, and show the training curve in Fig. 5. Compared with HER that is trained using a normal policy, we observe that UDPO achieves similar performance, and sometimes it behaves better efficiency. For comparing UDPO with HRL methods, we also include HIGL on these source tasks. To our surprise, HIGL performs badly or even fails in many tasks, indicating its sensitivity by their landmarks sampling strategy. In Appendix C.1 we further illustrate the MSE error between the planned state and the real state that the agent actually achieved, with the visualization of the planning states, demonstrating that UDPO has a great planning ability under goal-conditioned challenges.

**Few-shot transferring to high-dimensional action spaces.** The `High-Dim-Action` transferring challenge requires the agent to generalize its state planner to various action spaces and action dynamics. In our design, the target task has higher action dimension with different dynamics (see Appendix B.2 for details), making the task hard to learn from scratch. As shown in Fig. 5, on the target *Fetch-Reach* and *Fetch-Push*, HER is much more struggle to learn well as it is in the source tasks, and all GCRL, HRL and contrastive RL baselines can only learn with much more samples or even fail. For PILoT, since the source and the target tasks share the same state space and state transition, we can transfer the goal-conditioned state planner and only have to train the inverse dynamics from scratch. In result, by simply augmented HER with additional transferring reward, HER (PILoT) shows impressive efficiency advantage for transferring the shared knowledge to new tasks.

**Few-shot transfer from vector to image states.** The `Vec-to-Image` transferring challenge learns high-dimensional visual observation input based agents guided by the planned goals learned from the low-level vector input. From Fig. 5, we can also learn that using the transferring reward of PILoT, HER (PILoT) achieves the best efficiency and final performance with much less samples on the two given tasks, compared with various baselines that are designed with complicated techniques.

**Few-shot transferring to different morphology.** We further test the `Complex-Morph` transferring challenge that requires to distill the locomotion knowledge from a simple Point robot to a much more complex Ant robot. The learning results from Fig. 5 again indicate impressive performance of HER (PILoT), while we surprisingly find that MEGA, HIGL and contrastive RL all can hardly learn feasible solutions on this task, even worse than HER. In our further comparison, we find that this tested task is actually more hard to determine the success by requiring the agent to reach at a very close distance (*i.e.*, less than 0.1, which can be further referred to Appendix B). In addition, the reason why MEGA fails to reach a good performance like the other tasks can be attribute to its dependency on an exploration strategy, which always chooses rarely achieved goals measured by its lowest density. This helps a lot in exploration when and the target goals' distribution is dense, like *Ant-Maze*. However, when the goals are scattered as in *Ant-Locomotion*, the agent has to explore a wide range of the goal space, which may lead MEGA's exploration strategy to be inefficient. On comparison, PILoT shows that, despite the task is difficult or the target goals are hard-explored, as long as we can transfer necessary knowledge from similar source tasks, agents can learn the skills quickly in few interactions.

**Zero-shot transfer for different layouts.** Finally, as we find the intermediate goals provided by the goal planner is accurate enough for every step that the agent encounters, we turn to a intuitive and interesting zero-shot knowledge transfer experiment for different map layouts. Specifically, we try to learn the solution on *Ant-Maze*, as shown in Fig. 4, which is a hard-exploring tasks since the wall between the starting point and the target position requires the agent to first increase the distance to the goal before finally reaching it. As Fig. 6 illustrates, simply deploying HER fails. With sufficient exploration, all recent GCRL, HRL and contrastive RL baselines can learn a feasible solution after a quite large times of sampling. However, since PILoT provide a way of distilling the goal transition knowledge from a much simpler tasks, *i.e.*, the *2D-Maze* task, we take the

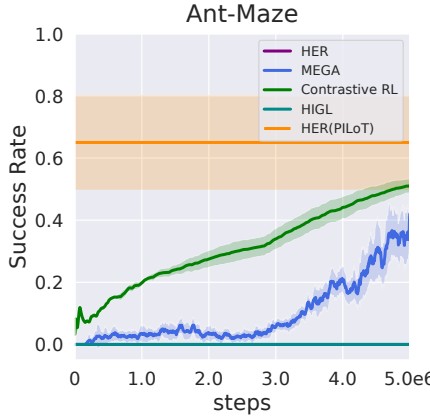

Figure 6: Training and zero-shot transferring curves on Ant-Maze task across 10 seeds.

intermediate goals as short guidance for the Ant-Locomotion policy pre-trained in Fig. 5. Note that the Ant-Locomotion policy provides the ability to reach arbitrary goals within the map. In this way, PILoT achieves zero-shot transfer performance without any sampling. This shows a promising disentanglement of goal planner and motion controler for resolving complex tasks.

## 7 CONCLUSION AND LIMITATION

In this paper, we provide a general solution for skill transferring across various agents. In particular, We propose PILoT, *i.e.*, Planning Immediate Landmarks of Targets. First, PILoT utilizes and extends a decoupled policy structure to learn a goal-conditioned state planner by universal decoupled policy optimization; then, a goal-planner is distilled to plan immediate landmarks in a model-free style that can be shared among different agents. To validate our proposal, we further design kinds of transferring challenges and show different usages of PILoT, such as few-shot transferring across different action spaces and dynamics, from low-dimensional vector states to image inputs, from simple robot to complicated morphology; we also show a promising case of zero-shot transferring on the harder *Ant-Maze* task. However, we find that the proposed PILoT solution mainly limited in those tasks that have clear goal transitions that can be easily distilled, such as navigation tasks; on the contrary, for those tasks which takes the positions of objects as goals, it will be much harder to transfer the knowledge since the goals are always static when agents do not touch the objects. We leave those kinds of tasks as future works.

## 8 ETHICS STATEMENT

This submission does not violate any ethics concern and adheres and acknowledge the ICLR Code of Ethics.

## 9 REPRODUCIBILITY STATEMENT

All experiments included in this paper are conducted over 5 random seeds for stability and reliability. The algorithm outline is included in Appendix A, the hyperparameters are included in Appendix B.4 and the implementation details are include in Appendix B.3. We promise to release our code to public after publication.

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

# Appendices

## A   ALGORITHM OUTLINE

---

**Algorithm 1** Planning Immediate Landmarks of Targets (PILoT)

---

1: **Source Task Input:** Empty replay buffer $\mathcal{B}_s$, state planner $h_\psi$, inverse dynamics model $I_\phi$ and goal planner $f_\omega$.
2: **Target Task Input:** Empty replay buffer $\mathcal{B}_t$, goal planner $f_\omega$, policy $\pi_\theta$.
   ▷ *Pre-training stage* trains UDPO on **source tasks**.
3: **for** $k = 0, 1, 2, \cdots$ **do**
4:     Collect trajectories $\{(s, a, s', g^t, r, \text{done})\}$ using current policy $\pi = \mathbb{E}_{\epsilon \sim \mathcal{N}}\left[I_\phi(a|s, h_\psi(\epsilon; s))\right]$ and store in $\mathcal{B}$
5:     Sample $(s, a, s', g_t, r) \sim \mathcal{B}_s$
6:     **if** *learn inverse dynamics function* **then**
7:       **repeat**
8:         Update $\phi$ by $L^I$ (Eq. (5))
9:       **until** Converged
10:     **end if**
   ▷ Distillation stage: distill goal planner from state planner.
11:     Update $\omega$ by $\nabla_\omega \mathcal{L}^f$ (Eq. (9))
12: **end for**
   ▷ *Transfer stage* trains HER (PILoT) on **target tasks**.
13: **for** $k = 0, 1, 2, \cdots$ **do**
14:     Collect trajectories $\{(s, a, s', g^t, r, \text{done})\}$ using current policy $\pi_\theta$
15:     Supplement the reward with additional bonus following Eq. (10):

$$r = r + r(s, a, \hat{g}'), \text{ where } \hat{g}' \sim f_\omega(\hat{g}'|\phi(s), g^t)$$

16:     Store $\{(s, a, s', g^t, r, \text{done})\}$ in $\mathcal{B}_t$
17:     Sample $\{(s, a, s', g^t, r, \text{done})\} \sim \mathcal{B}_t$
18:     Learn $\pi_\theta$ by HER
19: **end for**

---

## B   EXPERIMENT SETTINGS

### B.1   ENVIRONMENTS

We list important features of the tested environments as in Tab. 1. Note that the *Goal Reaching Distance* is rather important to decide the difficulty of the tasks, so we carefully choose them to meet the most of the current works.

| Environment Name | Obs. Type | Obs. Dim | Act. Dim | Goal Dim | Goal Reaching Distance | Episode Length |
|---|---|---|---|---|---|---|
| Fetch-Reach | Vector | 10 | 4 | 3 | 0.05 | 50 |
| Fetch-Push | Vector | 25 | 4 | 3 | 0.05 | 50 |
| Fetch-Reach-High-Dim | Vector | 10 | 8 | 3 | 0.05 | 50 |
| Fetch-Push-High-Dim | Vector | 25 | 8 | 3 | 0.05 | 50 |
| Fetch-Reach-Image | Image | 64*64*3 | 4 | 3 | 0.05 | 50 |
| Reacher | Vector | 11 | 3 | 2 | 0.02 | 50 |
| Reacher-Image | Image | 64*64*3 | 3 | 2 | 0.02 | 50 |
| Point-Locomotion | Vector | 6 | 2 | 2 | 0.1 | 500 |
| Ant-Locomotion | Vector | 29 | 8 | 2 | 0.1 | 100 |
| 2D-Maze | Vector | 2 | 2 | 2 | 1.0 | 50 |
| Ant-Maze | Vector | 29 | 8 | 2 | 1.0 | 500 |

Table 1: The environments used in our experiments, where goal reaching distance denotes the range of the goal; in other words, when the agent get closer to the desired goal than the distance, the task is regarded success.

It's worth noted that Ant robot in the common used *Ant-Maze* environment (*e.g.*, the one used in Pitis et al. (2020); Eysenbach et al. (2022)) is different the one in *Ant-Locomotion* (*e.g.*, the one used in Zhu et al. (2021)), like the *gear* and *ctrlrange* attributes. Thus, in order to test the transferring ability, we synchronize the Ant robot in these two tasks and re-run all baseline methods on these tasks.

## B.2 ACTION DYNAMICS SETTING FOR HIGH-DIM-ACT CHALLENGE

For transferring experiments on `High-Dim-Act` challenge, we take an 80% of the original gravity with a designed complicated dynamics for the transferring experiment (different both action space and dynamics). Particularly, given the original action space dimension $m$ and dynamics $s' = f_s(a)$ on state $s$, the new action dimension and dynamics become $n = 2m$ and $s' = f_s(h(a))$, where $h$ is constructed as:

$$h = -\exp(a[0:n/2]+1) + \exp(a[n/2:-1]))/1.5$$

here $a[i:j]$ selects the $i$-th to $(j-1)$-th elements from the action vector $a$. In other words, we transfer to a different gravity setting while doubling the action space and construct a more complicated action dynamics for agent to learn.

## B.3 IMPLEMENTATION DETAILS

The implementation of PILoT and HER are based on a open-source Pytorch code framework[1]. As for compared baselines, we take their official implementation, use its default hyperparameters and try our best to tune important ones:

- MEGA (Pitis et al., 2020): `https://github.com/spitis/mrl`
- HIGL (Kim et al., 2021): `https://github.com/junsu-kim97/HIGL`
- Contrastive RL (Eysenbach et al., 2022): `https://github.com/google-research/google-research/tree/master/contrastive_rl`

For resolving image-based tasks, we learn an encoder that is shared between the policy and the critic. In particular, we use the same encoder structure for HER, MEGA and HIGL. The encoder has four convolution layers with the same $3 \times 3$ kernel size and 32 output channel. The stride of the first layer is 2 and the stride of the other layers is 1, as shown in Fig. 7. We adopt ReLU as the activation function in all the layers. After convolution, we have a fully connected layer with 50 hidden units and a layer-norm layer to get the output of the encoder. When training, only the gradients from Q network are used to update the encoder. For contrastive RL, we take their default structures for image-based tasks.

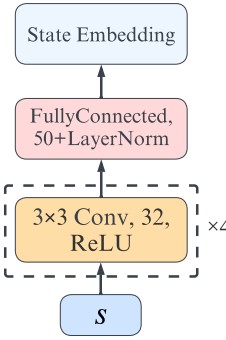

Figure 7: The encoder network architecture.

## B.4 IMPORTANT HYPERPARAMETERS

We list the key hyperparameters of the best performance of HER in Tab. 2 and UDPO in Tab. 3 on each source task. For each task, we first tune HER to achieve the best performance, based on which we further slightly adjust UDPO's additional hyperparameters.

For HER, we tune *Replay buffer size* $\in \{1e5, 1e6\}$, *Batch size* $\in \{128, 2048, 4096\}$, *Policy $\pi$ learning rate* $\in \{3e-4, 1e-3\}$.

For UDPO, we only tune two hyperparameters, *State planner coefficient* $\lambda \in \{1e-3, 5e-3, 1e-2, 5e-2, 1e-1\}$, *Inverse dynamics $I$ learning interval* $\Delta \in \{200, 500, 1500, 2000\}$. As further shown in Section C.2, the choice slightly affects the success rate but can impose considerable influence on the accuracy of the state planner. The larger $\lambda$ will lead stronger constraint on the accuracy

---

[1] `https://github.com/Ericonaldo/ILSwiss`

of the state planner, but can hurt the exploration. On the other hand, $\Delta$ controls the training stability, and a larger $\Delta$ assumes that the local inverse dynamics does not change for a longer time. Therefore, in principle, for those tasks that exploration is much more difficult, we tend to choose a small $\lambda$; for those tasks that the algorithm can learn fast so that the local inverse dynamics changes drastically, we should have a small $\Delta$. In default, we choose $\lambda = 1e - 2$ and $\Delta = 1500$.

We also list the hyperparameters of HER (PILoT) in Tab. 4 on each target task, which is the same as the baseline HER algorithm (except the additional *Transferring bonus rate*.) In default, we set all *Transferring bonus rate* to be 1.0, and find that it can reach a desired performance. In Section C.2, we also include the ablation of the choice of this hyperparameter.

Table 2: Hyperparameters of HER on the source tasks.

| Environments | Fetch-Reach | Fetch-Push | Reacher | 2D Maze | Point-Locomotion |
|---|---|---|---|---|---|
| Optimizer | Adam Optimizer | | | | |
| Discount factor $\gamma$ | 0.99 | | | | |
| Replay buffer size | 1e5 | 1e6 | 1e6 | 1e5 | 1e6 |
| Batch size | 128 | 2048 | 2048 | 128 | 2048 |
| Number of VecEnvs | 1 | 8 | 4 | 1 | 4 |
| $Q$ learning rate | 3e-4 | | | | |
| Policy $\pi$ learning rate | 3e-4 | 1e-3 | | 3e-4 | |

Table 3: Hyperparameters of UDPO on the source tasks.

| Environments | Fetch-Reach | Fetch-Push | Reacher | 2D Maze | Point-Locomotion |
|---|---|---|---|---|---|
| Optimizer | Adam Optimizer | | | | |
| Discount factor $\gamma$ | 0.99 | | | | |
| Replay buffer size | 1e5 | 1e6 | 1e6 | 1e5 | 1e6 |
| Batch size | 128 | 2048 | 2048 | 128 | 2048 |
| Number of VecEnvs | 1 | 8 | 4 | 1 | 4 |
| State planner coefficient $\lambda$ | 1e-2 | | | 1e-1 | 5e-3 |
| $Q$ learning rate | 3e-4 | | | | |
| Policy $\pi$ learning rate | 3e-4 | 1e-3 | | 3e-4 | |
| Inverse dynamics $I$ learning rate | 1e-4 | | | | |
| Inverse dynamics $I$ learning interval $\Delta$ (epochs) | 200 | | | 1500 | |

Table 4: Hyperparameters of HER / HER (PILoT) on the target tasks.

| Environments | Fetch-Reach-High-Dim | Fetch-Push-High-Dim | Fetch-Reach-Image | Reacher-Image | Ant-Locomotion |
|---|---|---|---|---|---|
| Optimizer | Adam Optimizer | | | | |
| Discount factor $\gamma$ | 0.99 | | | | |
| Replay buffer size | 1e5 | 1e6 | | | |
| Batch size | 128 | 2048 | | | 4096 |
| Number of VecEnvs | 1 | 8 | 1 | 4 | 4 |
| $Q$ learning rate | 3e-4 | | | | |
| Policy $\pi$ learning rate | 3e-4 | 1e-3 | 3e-4 | 1e-3 | 1e-3 |
| Transferring bonus rate | - | - | 1.0 | | |

# C  MORE EXPERIMENTAL RESULTS

## C.1  DOES UDPO REACH WHERE IT PREDICTS?

In the universal decoupled policy structure, the state planner is decoupled and trained for predicting the future plans that the agent is required to reach. Therefore, it is critical to understand the plans given by the planner and make sure it is accurate enough that the agent can reach where it plans to go, for distilling and transferring. To this end, we analyze the distance of the reaching states and the predicted consecutive states and draw the mean square error (MSE) along the RL leaning procedure in Fig. 8. To our delight, as the training goes, the gap between the planned states and the achieved states is becoming smaller, indicating the accuracy of the state plan.

Additionally, we also visualize the imagined rollout by state planner on the source tasks, which is generated by consecutively take a predicted states as a new input. We compare it with the real rollout in Fig. 9, showing the state planner can conduct reasonable multi-step plan.

On the target tasks, we visualize the subgoals proposed by the distilled goal planner and the real rollout that was achieved during the interaction in Fig. 10, showing the effective and explainable guidance from the goal planner.

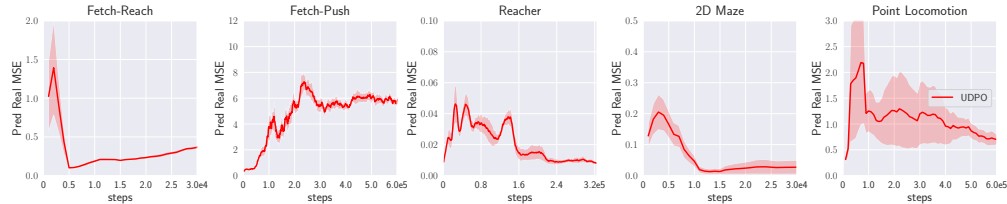

Figure 8: Curves of the MSE between one-step prediction of the state planner and the real state achieved in the source environments.

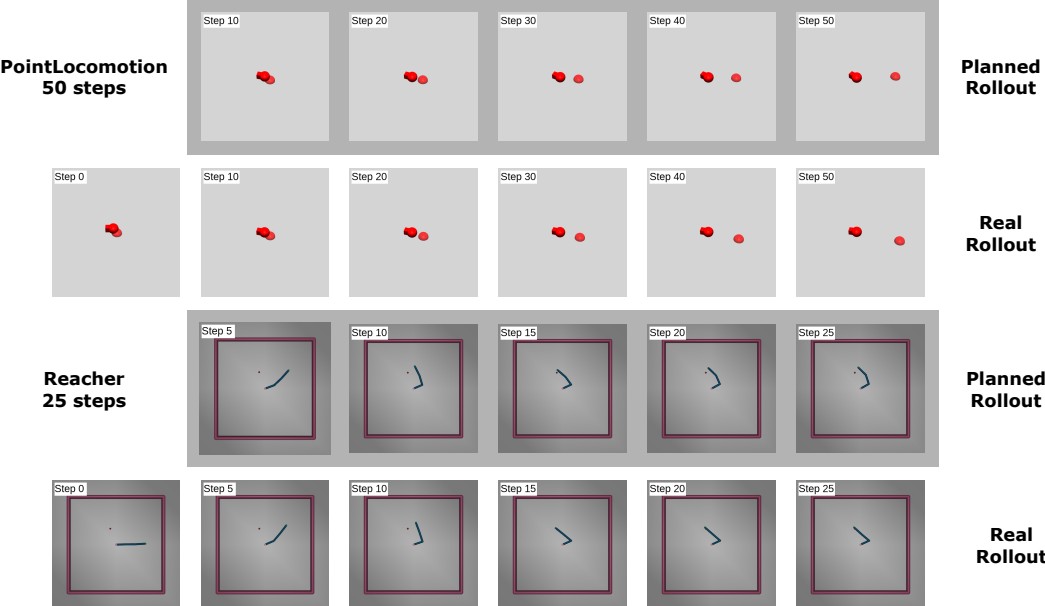

Figure 9: Imagined rollout by state planner and the real rollout that was achieved during the interaction in two source tasks, showing the reasonable multi-step plan of the state planner. For Fetch-Reach and Fetch-Push environments, we cannot render the planned rollout since the simulator internal states can not be recovered from agent observations.

## C.2 ABLATION STUDY

In this section, we aim to investigate the robustness and the key components of our proposed PILoT framework. Specifically, we first analyse the two critical hyperparameters, *i.e.*, the inverse dynamic train frequency $\Delta$ and the regularization coefficient $\lambda$ on training UDPO in the pre-training stage; then, we provide additional ablation analysis on bonus ratio $\beta$ in the transferring stage.

**Pre-training ablation on the inverse dynamic train frequency $\Delta$.** We first conduct ablation studies on the inverse dynamic train frequency $\Delta$. In particular, this hyperparameter determines how often we train the low-level inverse dynamics and how long we regard the inverse dynamics is static when we train the high-level state planner. It is intuitive that a larger $\Delta$ is assuming that the local inverse dynamics does not change for a longer time; on the contrary, a small $\Delta$ should be used when the the local inverse dynamics changes drastically. In our experiments, we find that $\Delta$ affects

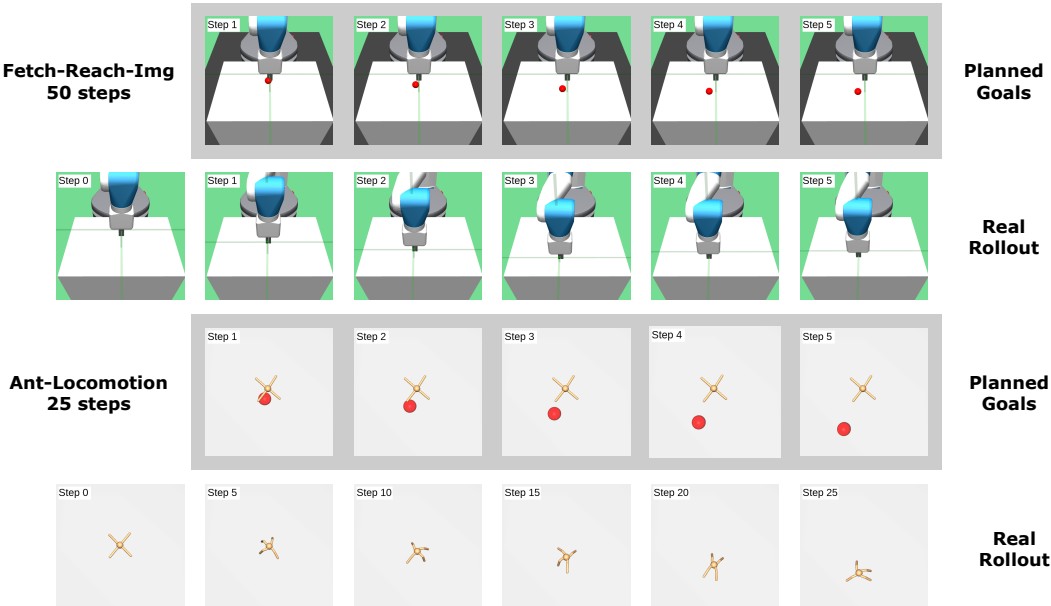

Figure 10: Subgoals proposed by the distilled goal planner and the real rollout that was achieved during the interaction in two target tasks, showing the effective and explainable guidance from the goal planner. For Ant-Locomotion environment, the goal planner is distilled from Point-Locomotion environment, in which the point agent has faster moving speed than the ant. Nevertheless, the resultant trajectory in Ant-Locomotion can still match the planned goals roughly.

the training stability, as it embeds some prior about the policy training in certain tasks. As is shown in Fig. 11, *Fetch-Push* requires smaller $\Delta$ than *Point-Locomotion*, since *Fetch-Push* is more simple, and the policy training is much faster than the policy training in the *Point-Locomotion*.

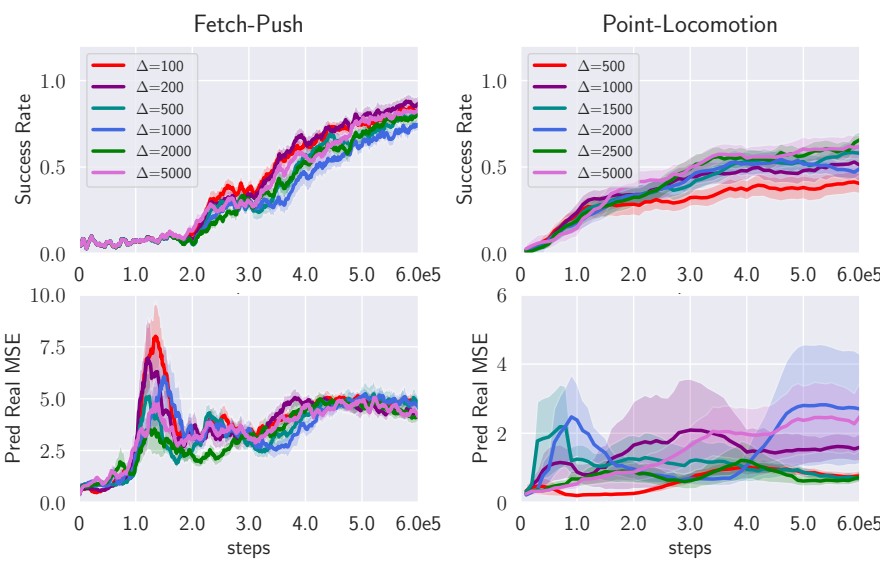

Figure 11: Ablation study on inverse dynamic train frequency $\Delta$ with 5 seeds, given the other parameters the same in Section 6.2.

**Pre-training ablation on the regularization coefficient $\lambda$.** The regularization coefficient $\lambda$ is another critical hyperparameter in UDPO training. The choice of $\lambda$ balances the policy gradient term and the constraint term in state planning updates. Particularly, larger $\lambda$ puts more weight on the

supervised penalty objective, which reduces the planning of infeasible next states. However this can hurt the exploration ability offered by policy gradient objective. The results in Fig. 12 supports the intuition. In both environments, the models trained with $\lambda = 0.1$ performs best in reaching where they plan, but can not finish the goal-reaching task well. On the other hand, a extreme small $\lambda$ results in a quite large gap between reached states and planned states. Such gap can make the subsequent transferring impossible. Therefore, the recipe is to find the medium $\lambda$ which achieves competitive success rate while keeping the value of prediction real MSE from explosion.

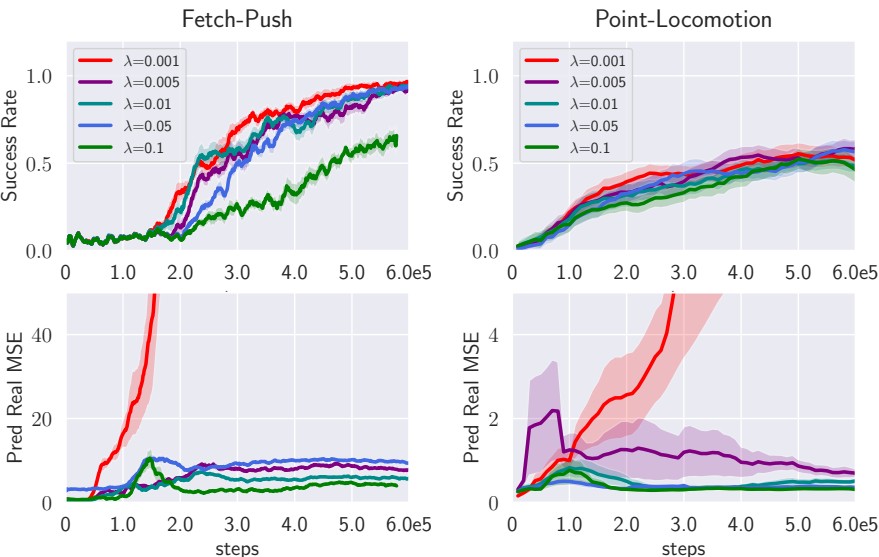

Figure 12: Ablation study on state planner coefficient $\lambda$ with 5 seeds, given the other parameters the same in Section 6.2.

**Transferring ablation on the bonus ratio $\beta$.** In the transferring stage, bonus ratio $\beta$ balances the similarity reward from distilled planner and the sparse reward from the environment. We observe from Fig. 13 that 1.0 is an appropriate choice for at least tasks tested in this paper. When $\beta$ is smaller (*e.g.*, 0.1, 0.2, 0.5), the success rate converges more slowly, and the final performance is also worse. This indicates that small $\beta$ can not provide strong enough signals for the policy to follow planned landmarks, leading to a decreased transfer efficiency. On the other hand, when a much larger $\beta$ (*e.g.*, 2.0, 5.0) is adopted, the performance becomes even worse than insufficient guidance from small $\beta$. Also, in Fetch-Reach-Image environment, the training curves are quite unstable. Although the planned landmarks are useful in overcoming the sparse reward issue, they can not completely replace the final sparse reward. As Fig. 8 shows, even in the source environments, there exist small but not negligible errors between planned states and achieved states. Using large $\beta$ can let the policy misled by the goal planner and overfitted to those errors.

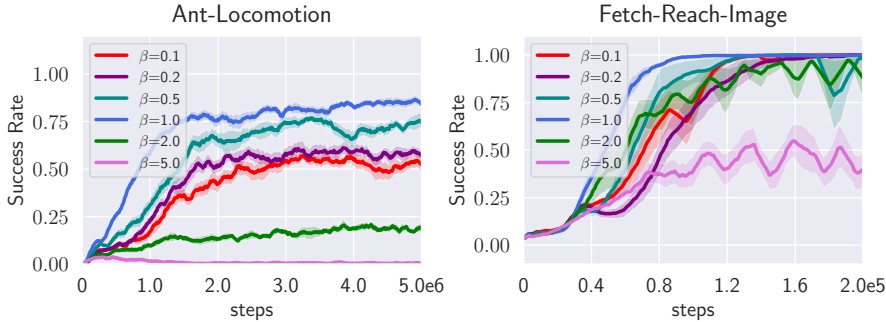

Figure 13: Ablation study on bonus ratio $\beta$ with 5 seeds, given the other parameters the same in Section 6.2.

## C.3   MORE ZERO-SHOT TRANSFER VISUALIZATION

In this section we illustrate more zero-shot transfer cases including success cases and failure cases. In fact, the failures should be attributed to the inaccuracy of the controller (policy) since the goal planner always gives the right way to success. If we can train a more accurate local controller, the performance of success rate can no doubt be further improved.

### C.3.1   SUCCESS CASES

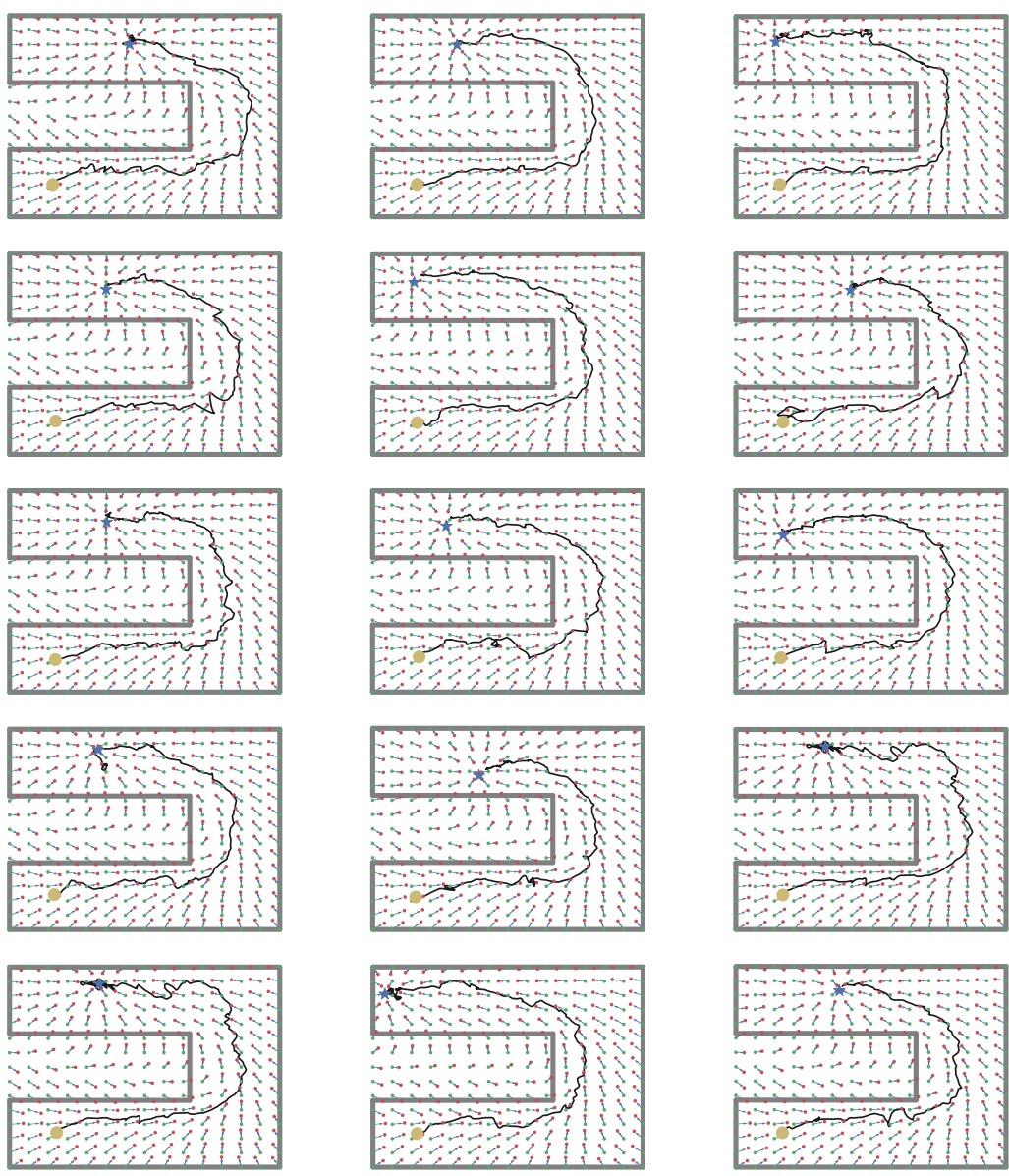

Figure 14: Success cases of zero-shot transfer on Ant-Maze.

## C.3.2 FAILURE CASES

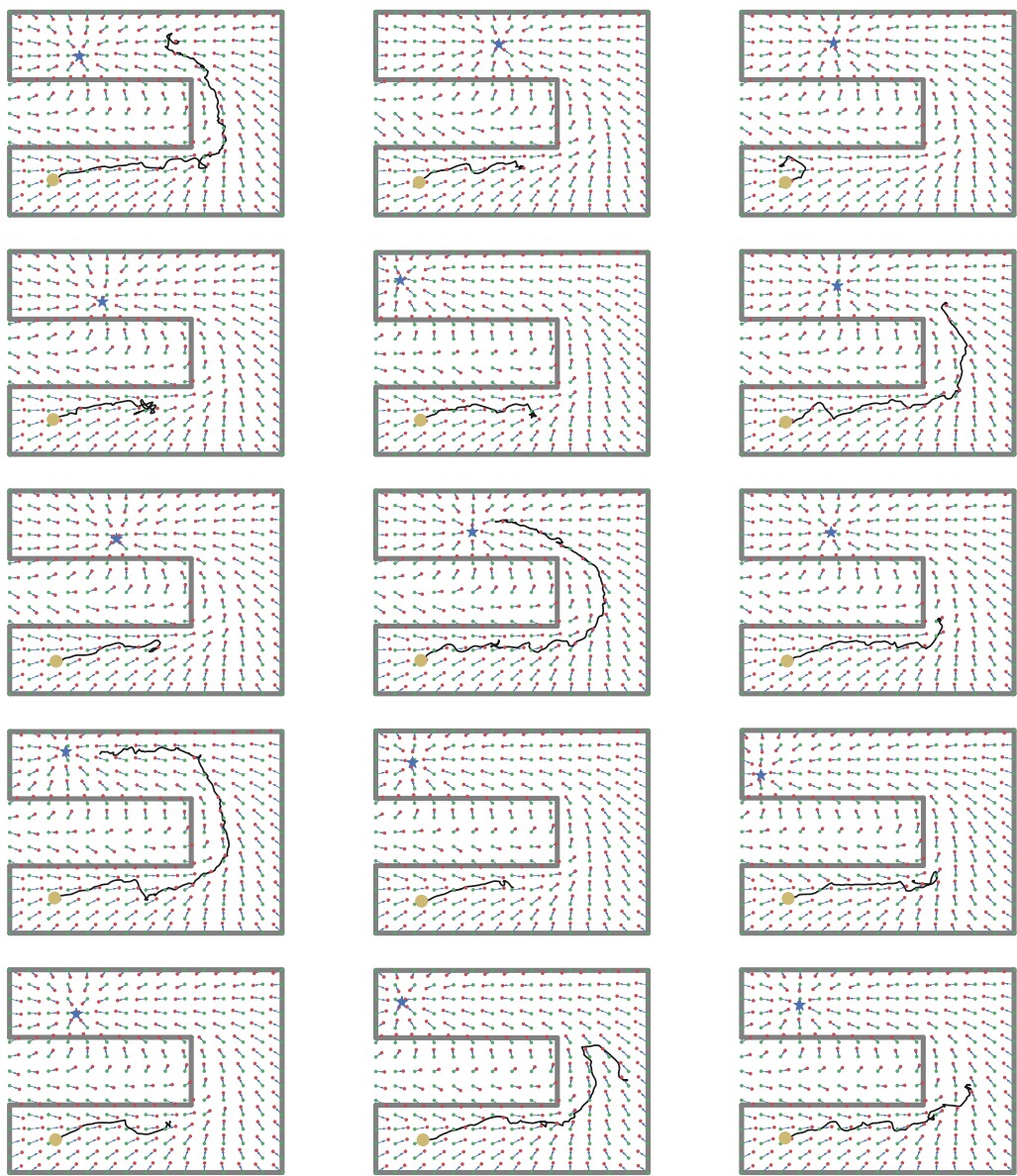

Figure 15: Failure cases of zero-shot transfer on Ant-Maze.

