# OpenReview forum: "Planning Immediate Landmarks of Targets for Model-Free Skill Transfer across Agents"
_ICLR.cc/2023/Conference — Submitted to ICLR 2023_

### Official Review · Reviewer_1XEZ · 2022-10-26

**Confidence:** 3
**Correctness:** 3
**Technical Novelty And Significance:** 3
**Empirical Novelty And Significance:** 2
**Recommendation:** 5

**Clarity, Quality, Novelty And Reproducibility:**

The proposed method is comprised of three stages, the latter two of which are original, while the first is based on prior work by Liu et al., 2022b, extended to the goal-conditioned setting. The submission is organized logically, and generally has the necessary experiments and ablations for a high-quality submission, excluding the listed components in Weakness (2) and (3). The clarity of the manuscript can be improved at the word-level by fixing the various typos and grammatical errors present in the submission, but other than these mistakes, the experimental settings are clearly detailed (after reading the full paper and appendix).

**Strength And Weaknesses:**

Strengths Of The Paper:
1) The paper generalizes the Decoupled Policy Gradient, which has been shown to be effective in the standard RL setting, to a goal-conditioned multi-task setting.
2) The proposed method is appealing because it facilitates building an informative dense reward in domains where it may otherwise be hard to construct one, such as for visual reinforcement learning tasks, where estimating the pose of the agent (for pose-based goals) is challenging.

3) Experiments are generally of high-quality, including proper documentation of hyperparameters, a description for how tuning was performed, and an appropriate set of ablations (which are in the Appendix, not the main body of the paper).
4) Seven tasks are used for evaluation, which is an appropriate number.

Weaknesses Of The Paper:

1) The first main weakness of the paper is the empirical results. Particularly, in Figure 5, Page 8, the complete method (UDPO) is compared against HER and HIGL, two baseline goal-conditioned reinforcement learning approaches, and the results do not support the authors’ claim that “Compared with HER that is trained using a normal policy, we observe that UDPO
achieves similar performance, and sometimes it behaves better efficiency.” On three of five source tasks, HER appears to outperform UDPO. On one of five, performance is nearly the same, and on the last (the 2D-Maze pointmass environment), UDPO outperforms HER. This may suggest UDPO incurs a penalty on the source-task it is initially trained. Transfer results are more compelling, where HER with the PILoT reward bonus is the best on four of five tasks.

2) For zero-shot experiments, where a task-specific planner is used with a pretrained lower-level on the AntMaze task (see Figure 6), more appropriate baselines for this experiment would be Planning with Goal-Conditioned Policies (Nasiriany, et al. 2019) and Hierarchically Decoupled Imitation for Morphological Transfer (Hejna III, et al. 2020). These suggestions are RL algorithms designed for zero-shot transfer of lower-level skills to new tasks and morphologies, and they operate with similar assumptions as this paper (that one has a pre-trained goal-conditioned lower-level policy in the target domain to be adapted to the new task).

3) Figure 5 is missing a comparison against other intrinsic reward methods. Currently, this figure shows that the combination of the informative goals sampled from the shared planner, and the bonus reward computed from these goals improves efficiency. However, it remains unclear how the proposed intrinsic reward compares to existing techniques for densifying sparse rewards. This is a particularly important evaluation to include because Figure 13, Page 18 suggests that the majority of the performance gains are due to the reward bonus (see $\beta = 0.1$ results).

Suggestions For The Paper:

* Agent-Agnostic Reinforcement Learning with Transformers is discussed in the related works, and Gupta et al. (2021) is not the only, nor the most recent paper that explores this. Consider referencing My Body is a Cage: the Role of Morphology in Graph-Based Incompatible Control, (Kurin et al. 2020) and AnyMorph: Learning Transferable Policies By Inferring Agent Morphology, (Trabucco et al. 2022), two other recent papers in this domain.

* There are a few typos, such as “Euclid distance” on Page 5, which is likely intended to read “Euclidean distance” instead. Fixing these will improve the readability of the manuscript.


**Summary Of The Paper:**

This paper investigates the subject of few-shot and zero-shot skill transfer between tasks and agent morphologies, two important multi-task reinforcement learning settings. The paper proposes a three-stage approach that first trains a goal-conditioned state-based planner in a source domain that is trained to sample next states that, when passed through an inverse dynamics model, lead to optimal actions. In this first stage, the composition of the one-step planning model and the inverse dynamics model is a policy for the source domain. This first stage is based on prior work by Liu et al., 2022b, but is extended by the authors to a goal-conditioned setting, whereas it was originally used by Liu et al., 2022b without goals.

This modularization of the policy is important to the authors’ method, as it can be distilled into a shared latent (goal) space that lends itself to transfer between different agents and tasks. This contribution, ie distilling a one-step planning model into a shared latent (goal) space is novel, though it is not clear why this is advantageous over directly learning this one-step model in the shared latent (goal) space in the first stage, instead of performing distillation post-hoc.

In the final stage of the method, a policy is learned in a target domain using goals specified in the shared latent (goal) space from stage two, and a reward bonus derived from the goals sampled from from the distilled one-step planning model (to mitigate reward sparsity). The advantage of the proposed method is that the combination of informative goals and a dense reward can improve sample efficiency. Overall, the proposed method is complex, illustrated by the three distinct stages of training required to learn a policy in the target domain. Evaluation is performed on seven distinct simulated environments; however, the results are somewhat mixed.


**Summary Of The Review:**

Overall, this paper proposes a surprisingly complex method in light of the gains observed. The weaknesses of the paper only slightly out-weigh its strengths, and once these weaknesses are resolved, I am willing to reconsider my evaluation and increase my review score. However, in light of these weaknesses, I am currently unable to recommend the paper for acceptance, and encourage the authors to discuss my questions and contribute new results in their rebuttal.

---

### Official Review · Reviewer_7PL4 · 2022-10-27

**Confidence:** 3
**Correctness:** 3
**Technical Novelty And Significance:** 2
**Empirical Novelty And Significance:** 2
**Recommendation:** 3

**Clarity, Quality, Novelty And Reproducibility:**

Clarity

The paper is clear about its contributions and well structured.

Quality

The paper is rather poorly written, with many typos and confusingly worded sections.

Originality

The method presented in this paper is novel and represents a clear but incremental improvement over prior work.

**Strength And Weaknesses:**

Strengths:
1) The paper is well structured and clear about its goals, contributions and placement compared to related work.

2) The method makes intuitive sense and could be extended to many interesting tasks. In the tasks presented it represents a clear improvement over prior methods.

Weaknesses:

1) There are many typos and confusingly worded sections that make the paper difficult to understand. The authors should carefully proofread and correct the incomplete sentences and other grammatical issues.

2) The methods in this paper are closely related to DePO (Liu et al., 2022b) and the method proposed (Sun et al., 2022). It would be helpful to compare against these two methods when it is applicable, such as when transferring from vector to image based state representation, and in the zero-shot experiment. The environments used in the paper are also quite simple. I would like to see results on some of the environments used in prior works, such as Sawyer Bin and PointSpiral from Contrastive RL (Eysenbach et al., 2022) or PointMaze and FetchPickAndPlace from Maximum Entropy Gain Exploration (Pitis et al., 2020), which were both used as baseline algorithms in this paper.

3) Although this is a transfer learning method, it seems that every each source task is only used to transfer knowledge to one target task. It would be nice to see one distilled goal planner be used to speed learning on several tasks. As it is now, it could be more fair to compare FLOPs used to train source + target policies to FLOPs used training directly on the target task.

Questions:

1) Is the sampling strategy $\pi_B$ in Section 4.1 always the goal conditioned policy being learned by DePO?

2) How is the embedding function $\phi$ learned for the target task?

3) Could the goal planner be more general if it is distilled from several state transition planners?

Minor:

1) Where is the small blue point in Figure 1?

2) Referring to the starting position (green points) as goals is confusing

3) Equation 2 uses notation $h_{\pi}(s^\prime | s)$ and $I(\cdot | s, s^\prime)$ from the DePO paper, but they are not defined until later

4) Figure 5 has five source and five target tasks

**Summary Of The Paper:**

This paper aims to achieve few or zero shot transfer pre-trained skills across tasks with different state and action spaces and environment dynamics, reducing the need for retraining.

The method involves 3 main steps. First, a goal-conditioned extension of DePO is used to learn an inverse dynamics model $I(a | s, s^\prime)$  and state transition planner $h_{\pi}(s^\prime | s, g^t)$. Next, the state transition planner, specific to the source task, is distilled into a general goal transition planner $f_\omega (g^\prime | g, g^t)$ by learning an embedding function $g = \phi(s)$ and using $h_\pi$ as a teacher. Finally, the goal planner is used to provide a dense reward $r(s, a, s^\prime, g^\prime) = cos_similarity(\phi(s^\prime), g^\prime)$ in the target task. Unlike prior methods, PILoT does not require the source and target tasks to have the same state or action space. The primary contribution of this paper is to switch from learning latent state representations to learning goal representations, which relieves the need for a shared action space and shared transition dynamics.

The dense rewards provided by the goal planner successfully help the agent learn the target task faster than several baseline methods, and to achieve zero-shot transfer learning from a point policy to an ant with a pretrained locomotion policy.

**Summary Of The Review:**

While the authors present some interesting ideas, the environments and baselines are lacking. Without additional experiments, it is difficult to determine whether this method represents an improvement over prior works. The writing could use some further proofreading and revision. Therefore, I am hesitant recommend this paper for acceptance as is, but would be happy to reconsider after some revision.

---

### Official Review · Reviewer_LhAd · 2022-10-28

**Confidence:** 4
**Correctness:** 2
**Technical Novelty And Significance:** 3
**Empirical Novelty And Significance:** 3
**Recommendation:** 5

**Clarity, Quality, Novelty And Reproducibility:**

Clarity: Main weakness of the paper. I really struggled to understand important aspects of the paper.

Quality: It would be valuable to understand why the baselines are chosen, and how the transfer tasks represent the main points that are being conveyed in the paper.

Novelty: The approach seems heavily based on Liu et al. (2022) with the addition of making the state planner goal conditional, and the distillation of this planner to transfer to new environments and agent settings. I think this is sufficiently novel and targets an important problem.

**Strength And Weaknesses:**


**Stengths:**
- Relevance:
	- The approach addresses a relevant problem which occurs in practice in multiple scenarios.
	- The policy distillation approach can be trained without experience in the target environment, since it is distilled from the expert policy, it assumes though that all the states correspond to subgoals (as per equation 10), or that the subgoals are known a priori. In the second case, this seems like a big assumption. In the first case, isn't this the same setting as the one in Figure 2a?
- Thorough ablations and analysis of experiments.
- The method manages to learn policies in the source domain competitively with other goal-conditioned methods, and shows better transfer properties in new domains.
- The method manages to transfer to new domains, new morphologies and observation spaces with minimal new data from the new domain, and it even can perform goal conditioned tasks in cases where the target and source domain have the same morphologies but different layouts.

**Weaknesses:**

- Method:
	- The method requires us to learn an inverse dynamics model for every new policy. Isn't it a big assumption that we can learn that for a wide enough coverage of states? Couldn't we just train a policy with the data used to learn the inverse dynamics model?
- Wrong claims and gaps in literature review:
	- It's not true that most of the work in RL is focused on a single task, single agent. There is extensive literature in methods [1,2] and benchmarks [2] to evaluate multi-task RL. There are also works [4] addressing agents that operate across different state and action spaces. Similarly, when introducing Decoupled Policy Optimization, see any method with successor representations for instance.
- Clarity:
	- The paper needs significant rewriting, the motivation in the introduction is confusing and difficult to follow. I would suggest starting with the SLAMcase study and why we care about similar goals even when the state and transitions are different.
	- I found the paper very difficult to read, specially the abstract and intro, which don't really talk about the main point of the paper, shown in Figure 2. The paper required me to keep checking the references (Sun et al, Liu et al.). It would help if the paper was a bit more self-contained and explained the important aspects of these references in a clearer way.
	- It would help in decoupled policy optimization if authors explained the terms in Eq. 2.
	- Similarly, it would help if the examples in section 3 for different assumptions were followed by some example of agent transfer that satisfies the different assumptions.
	- Figure 3 is not very informative, I would recommend specifying there what are the losses, what the colors represent and explaining the process more detailedly in the caption.
	- Experiments are also unclearly described.
	- It would be good if authors pointed more explicitly how their training method is akin to HER, which they only mention in the method section as a method that can be used, to later find in the experiment section that this is how it is trained.
	- I would suggest changing the name HER (PILOT and UDPO Pilot) and just call it under the same name, or the same name with an underscore original/transferred.
	- I would like some text describing the reason why the baselines are selected, what are the ideas that want to be tested when comparing with HER vs HIGL vs MEGA vs ContrastiveRL? Some of it inferred from the related work, but it would be good to reiterate on that when explaining the baselines.
	- In the zero-shot transfer experiment, how is the goal transition function learned? How can we learn from the Ant-locomotion, and without any retraining, that one cannot follow a straight line in ant-maze to complete a goal.


**Clarification Questions:**
- What does it mean in practice the equation in the end of page 2? What kind of environments satisfy this condition? There is also an abuse of notation, using T to indicate the state at the transition function and the likelihood of a state transition.

- How does the objective in Eq. 7 help? We are maximizing the planner to match the actions from the policy, but the policy is being learned based on this planner. When the policy is random at the beginning, how does this give information to h?



- The diagram c in figure 2 seems to imply that the requirement is that the transitions between subgoals is the same. What happens if there are two agents with similar subgoals but one can achieve it faster than the other? For instance, an agent that moves half the distance of the other agent. Would those two agents be transferable under this setting? I am not sure Figure 2. c represents that case. In fact, the transfer experiments work with different dynamics, which means that while both agents may have a set of shared subgoals, some of the intermediate states (or whatever projection for them) will be different, so I am not sure how the method would work here. For instance, if the dynamics change and suddenly an action moves the arm at a faster pace, some of the goal states in the original environment won't appear in the transfer environment.

**Summary Of The Paper:**

This paper proposes a method to transfer policies across agents having different observations and action spaces in a given environment, but a shared space of subgoals and the same set of subgoals that lead to a given goal. To do so, it trains a policy in an origin environment, decoupling it into an inverse dynamic model, and a goal-conditioned state planner. This decoupling allows to transfer to new agents by learning a new inverse dynamic model and distilling the goal-conditioned state planner from the original agent. The approach is tested in different benchmarks, changing the observation and action space between domains, dynamics, and layout. The proposed approach can learn in the new domains much faster than competing baselines, and even execute goal-conditioned policies in a zero-hot manner given changes in environment layouts.

**Summary Of The Review:**

My main concern is with the clarity of the paper. I believe I understand what the method is doing, but the assumptions  it relies on don't seem to match with the described experiments. There are a lot of missing details in the method section. I also struggle to understand the difference between Fig2 a and Fig2 c. Even if the ideas in this paper are interesting, and results promising, much of the paper is to unclear to be accepted at ICLR. I am giving the paper a weak reject, with the hope that the questions above will be clarified, and that the paper can be updated to clarify experiments and framing of the method.

---

> ### Author Response · Authors · 2022-12-12
> **Ask for references**
>
> Dear reviewer,
>
> It seems that you did not include your references in your review, would you mind supplement them so that we can make a better revision?
>
> Thanks!
>
> The authors

---

### Official Review · Reviewer_7J1u · 2022-10-28

**Confidence:** 4
**Correctness:** 3
**Technical Novelty And Significance:** 2
**Empirical Novelty And Significance:** 2
**Recommendation:** 3

**Clarity, Quality, Novelty And Reproducibility:**

The method section of the paper is reasonably easy to follow, and the claims seem technically sound. The issues with novelty and significance are as outlined above.

**Strength And Weaknesses:**

Strengths

1. The problem considered by the paper is important - enabling transfer of skills for agents between different tasks. This is because collecting samples is expensive for real systems, and efficiently reusing data can help the agent learn new useful tasks.

Weaknesses

1. Missing comparison to regular distillation

The main component of this approach that is used for transfer is the distillation between the state planners. However, distillation between policies for transfer is quite standard in the literature, especially for the case when the action space is the same, but only the observation space is different. This includes both the low-level state to high-dim images [1] and morphologically different robot [2] settings considered in this paper. It is possible that using the decoupled policy and distilling in just the state space will yield more robust transfer, but currently there are no experiments that examine /show this. For the other transfer setting considered in the paper (action space), the DePO [3] paper includes very similar results, establishing that the decoupled policy structure helps since only the inverse dynamics model needs to be trained, and the state planner can be reused.

2. Novelty/Significance

The paper largely builds on DePO, using the same decoupled policy structure and training objective formulation. The only changes are that goal conditioning is added to the policy, and there is an extra step of distilling the state planners when trying to transfer to new tasks. However, the idea of distillation for transfer is also quite standard as described in the point above [1-2]. Furthermore the experiments don't show transfer across different semantic tasks - i.e where the task is actually different (eg reaching to pushing etc), but rather for the same task, but just for different observation modalities or with different action space (the action dynamics setting does seem a little contrived as its not natural for the robot to suddenly have to deal with the kind of new actions considered in the paper). As discussed in the point above, it is known that student-teacher distillation helps for this kind of transfer (has also been used a lot for sim-real applications), and it's unclear if the decoupled policy structure adds any benefit to this application.


[1] Jain, Divye, et al. "Learning deep visuomotor policies for dexterous hand manipulation."
[2] Hejna, Donald, Lerrel Pinto, and Pieter Abbeel. "Hierarchically decoupled imitation for morphological transfer."
[3] Liu, Minghuan, et al. "Plan Your Target and Learn Your Skills: Transferable State-Only Imitation Learning via Decoupled Policy Optimization."


**Summary Of The Paper:**

The paper proposes an approach for transferring skills for goal-conditioned tasks with different action / observation spaces. The method seems to build on Decoupled Policy Optimization (DePO) to learn a planner for the next state, and an inverse dynamics model to produce the action, but additionally conditions on the desired goal. To transfer to a new task, a new goal planner is learned in the target domain via distillation (using the planner in the source domain as the teacher). Experimental domains include openai gym environments including fetch robot, ant-locomotion, reacher.

**Summary Of The Review:**

Overall I am not in favor of acceptance because the key idea of distillation for observation/action space transfer is well known in the literature, the decoupled policy architecture is from a previous paper[1], and the authors do not show that the decoupled architecture helps distillation, and so the contribution of the paper is unclear.

---

### Author Response · Authors · 2022-12-12
**Revising statement**

Dear reviewers,

Thanks for all of your detailed reviews, and we do find we need some rewrite to take all reviewers' suggestions to make a better work, which may lead to another round of submission.

Thanks again!

---

### Decision · Program_Chairs · 2023-01-20

**Decision:**

Reject

**Justification For Why Not Higher Score:**

No rebuttal.

**Justification For Why Not Lower Score:**

N/A

**Metareview: Summary, Strengths And Weaknesses:**

All reviews were concerned about the clarity of the draft, and also missing some critical experiments (e.g. ablation study of different components presented in the paper). The authors also did not submit any rebuttal, while several reviewers expressed that they were willing to discuss further. Hence, the intention of further improvement is unclear as well.